# Dynamics of Microbial Shedding in Market Pigs during Fasting and the Influence of Alginate Hydrogel Bead Supplementation during Transportation

**Mariana Fernandez [1], Arlene Garcia [1], David A. Vargas [2] and Alexandra Calle [1,***

1   School of Veterinary Medicine, Texas Tech University, Amarillo, TX 79106, USA;
    Mariana.fernandez@ttu.edu (M.F.); arlene.garcia@ttu.edu (A.G.)
2   Department of Animal and Food Sciences, Texas Tech University, Lubbock, TX 79409, USA;
    andres.vargas@ttu.edu
*   Correspondence: Alexandra.calle@ttu.edu; Tel.: +1-806-834-4074

**Abstract:** The shedding of foodborne pathogenic bacteria by food-animals can be affected by multiple factors, such as animal health, diet, stress, and environmental conditions. The practices that come with transport involve fasting, handling, mixing with unfamiliar pigs, and fluctuating temperatures. These practices, especially fasting and transport, can increase the microbial load in the feces of animals. The use of alginate hydrogels is a novel delivery system that can be a potential food safety intervention during transport to induce satiety and provide electrolytes to the animal's system. This study sought to observe microbial shedding as affected by fasting and hydrogel bead supplementation during transport. Sixty market pigs were subjected to a 12 h fasting period and an additional 4 h transport period, in which a treatment group was fed hydrogel beads and a control group was not. Sampling points were before fast (BF), before transport (BT), and after transport (AT). Fecal samples were collected from every animal at each sampling point. Results from this study showed a significant increase in the concentrations of both Enterobacteriaceae and *Escherichia coli* between the before fast (BF) and after transport (AT) sampling points. However, no difference ($p > 0.05$) was found between the treatment (hydrogel) and control (no hydrogel) during transport. Moreover, no significant difference was found in the prevalence of *Salmonella* and *E. coli* O157:H7 at the three different sampling points, or between the treatment and control groups.

**Keywords:** swine transport; feed withdrawal; pathogen shedding; stress; alginate hydrogel beads

## 1. Introduction

In food-animal production, animal health and wellness are interconnected with the pre-harvest interventions to ensure the safety of their products and to minimize the risks of foodborne illnesses to the consumers. The Centers for Disease Control and Prevention (CDC) estimates that pork is responsible for 525,000 foodborne infections, 2900 hospitalizations, and 82 deaths every year [1], which is in part due to the carriage of foodborne pathogens by swine [2]. The presence and concentration of these microorganisms in the gastrointestinal tract of pigs may vary due to numerous factors [3], such as sanitary conditions at the farm, quality and availability of water, quality of feed, and type of diet [4,5]. As a result, pre-harvest interventions are implemented prior to slaughter to minimize food safety risk factors and to aid with food safety of pork during processing [6].

The transportation of animals is essential to multisite pork production systems, since farrowing, finishing, and processing occur in different locations [7]. For pigs, transport is a complex stressor, as it involves handling, mixing of unfamiliar pigs, fluctuating temperatures, withdrawal from feed and water, sudden speed changes, and noise [8–10]. Early studies suggest that transport and the practices associated with it could lead to increased numbers of *Salmonella* in feces and the ileum, *E. coli* in feces, as well as increased prevalence of *Salmonella* [11–13]. Often, transportation and fasting alone do not affect shedding

rates of pathogens; however, the combination of fasting and transport, and the length of fasting can cause an increase in the levels of human pathogens, such as *Salmonella* and *Campylobacter jejuni* in the ileum and cecum, and the populations of Enterobacteriaceae in market pigs [14–16]. Feed withdrawal is currently used as a strategy to control and reduce carcass contamination [17]. Nevertheless, fasting for extended periods of times has been linked with increased foodborne pathogen colonization and a higher incidence of enteric infections in food-producing animals [15,18].

The implementation of practices to reduce pathogen shedding at the farm in animal production settings is part of "One Health" approach, which integrates the health of humans, animals, and their environment to prevent public health threats [19]. Thus, to ensure the safety of foods for human consumption, it is necessary to guarantee the health and welfare of animals [20]. Diet management is a common pre-harvest intervention [21]. A new strategy that has gained interest in the scientific community is the diet supplementation with hydrogels beads. They can be used to deliver bioactive compounds to animals for different purposes and could also help to control dehydration and hunger [22]. Hydrogels are a cross-linked three-dimensional network of structures, which consist of hydrophilic polymers [23]. The beads have a soft, rubbery consistency, with low interfacial tension with water or biological tissues [22]. This provides physical characteristics, which are similar to that of living tissue in the human body [24]. Supplementing alginate hydrogel beads with minerals has the potential to keep animals hydrated and full during transport. Additionally, with their viscosifying abilities, alginates have a potential to increase gastric distention and slow gastric emptying, which can result in satiety in animals [25]. Inducing satiety and controlling dehydration could have an effect in animal stress, which may potentially impact the rates of microbial shedding in the feces of swine during transport.

Considering the concern of pathogen increase associated with fasting, this study aimed to observe microbial shedding as affected by feed withdrawal prior to and during transportation. The specific objectives were (1) evaluate the changes in concentration of Enterobacteriaceae and *E. coli* shedding due to fasting and transportation (2) determine whether fasting and transportation influence the shedding of *Salmonella* and *E. coli* O157:H7, and (3) investigate the effect of hydrogel bead supplementation during transport on the concentration of Enterobacteriaceae and *E. coli* and pathogen shedding.

## 2. Materials and Methods

### 2.1. Swine, Pen Assignments, and Experimental Design

This study was conducted at the Texas Tech University New Deal Swine Unit located in Texas (referred to as "farm" in this document), between March and April. Sixty market-weight pigs (PIC Camborough) were used in this study. All animals were fed a diet to meet or exceed the National Research Council (NRC) nutritional requirements. Feed and water were provided ad libitum. Pigs were chosen for this study based on their weight (average: 100 kg) and sex (30 male; 30 female). Prior to the study, pigs were classified in equal numbers (20 animals/category) into three different weight categories: light (89–99 kg), medium (100–105 kg), and heavy (106–116 kg). Pigs were then randomly assigned to treatment and control groups. The treatment group included hydrogel bead supplementation during an acclimation period (one day prior to fasting) and during transport (TRT). The control group had no hydrogel bead supplementation during acclimation, nor during transport (CON). Once treatment and control groups were assigned, pigs were placed into either a TRT or CON pen based on their assigned treatment group weight category. Each pen was allotted with 6 pigs (3 male; 3 female).

### 2.2. Alginate Hydrogel Beads

Alginate hydrogel beads (referred to as hydrogel beads in this document) were made at the food chemistry laboratory in the Department of Animal and Food Sciences, Texas Tech University. The beads were prepared by mixing sodium alginate and calcium chlorate solutions; prepared separately and homogenized. The calcium chloride solution was mixed

with an electrolyte solution and homogenized. The hydrogel beads were approximately 1.5–2.0 cm in diameter. Immediately before the transport experiments, hydrogel beads were sprayed with a 1% swine fecal maternal pheromone solution, as an aid for acceptability by the pigs.

### 2.3. Transportation and Bead Supplementation

The pigs were fasted for 12 h, followed by a 4 h transport period meant to simulate transport conditions before slaughter. Two days prior to the study, only the treatment pigs were fed the hydrogel beads for acclimatization prior to transport. On the day of the study, pigs were transported in a two-compartment trailer that ensured the separation of the control and treatment groups. Wood shavings were used to bed the floor of the trailer. Hydrogel beads were only supplemented to the treatment group, by adding 1 kg/head along the perimeter of the treatment compartment. After the 4 h transportation period, pigs were returned to the farm and unloaded for sample collection.

### 2.4. Sample Collection

Fecal samples were collected from each animal at three pre-determined sampling points. The first time point was 24 h after the acclimation period, but 12 h before transport and prior to fasting; before fast (BF). After the BF sample, collection feed was removed from all pens. The second sample was collected before transport (BT), which corresponded to feces obtained after the 12 h fasting period. The third sample was obtained after transport (AT), upon the arrival of the pigs to the farm. To collect the samples, pigs were moved into a scale where fecal samples were collected. The samples were collected by using a sterile glove (new between each sample collection) and obtaining at least 10 g of fecal material directly from the rectum, which was then placed in pre-labeled sterile fecal collection cups and transported in a cooler with ice packs to the TTU Laboratory for further microbial analysis.

### 2.5. Microbial Analyses

#### 2.5.1. Enumeration of Microbial Indicators

Samples were processed at the food microbiology lab of Texas Tech University, Lubbock, TX, USA. For each sample, 10 g of feces were placed in a filtered Whirl-Pak® bag (Whirl-Pak®, Madison, WI, USA) with 90 mL of modified tryptic soy broth (mTSB) (Neogen Culture Media, Lansing, MI, USA), supplemented with 8 mg/L of novobiocin and acid digest of casein. This media was used as a diluent since it was needed for the further enrichment step to detect pathogens [26]. Samples were homogenized at 200 rpm for 2 min in a shaker (Thermo Scientific, MaxQ 2000, Waltham, MA, USA). Serial 10-fold dilutions were prepared by placing 1 mL of the homogenate in 9 mL tubes of buffered peptone water (BPW) (EDM, Darmstadt, Germany).

For the enumeration of Enterobacteriaceae, 1 mL of each dilution was plated in duplicate on 3M™ Petrifilm™ Enterobacteriaceae count plates (3M™, St. Paul, MN, USA). Plates were incubated for 24 h at 37 °C. For the enumeration of *E. coli*, 1 mL of each dilution was plated in duplicate on 3M™ Petrifilm™ *E. coli*/Coliform count plates (3M™, St. Paul, MN, USA). Plates were incubated for 48 h at 37 °C. Upon incubation times, colonies were enumerated, reported as CFU/g, and log transformed for statistical analysis.

#### 2.5.2. *Salmonella* and *E. coli* O157:H7 Detection

Sample homogenates initially diluted in mTSB were incubated for 24 h at 37 °C. Upon completion of incubation time, a second enrichment was performed by adding a 10 mL aliquot into 90 mL of mTSB. The second enrichment was performed to reduce the background microflora in the feces. The second enrichment was incubated in motion for 6 h at 37 °C. The samples were incubated in motion to allow for the nutrients in the media to evenly distribute and be available for bacteria [27]. Samples were screened for *Salmonella* and *E. coli* O157:H7 using the Real-Time BAX® System *Salmonella* assay

(BAX® System KIT2006, Hygiena, Camarillo, CA, USA) and the BAX® System Real-Time *E. coli* O157:H7 assay (BAX® System KIT 2000, Hygiena, Camarillo, CA, USA), respectively. The PCR assay was conducted following the manufacturer's protocol. In brief, 5 μL of each enriched sample was added to 200 μL of lysis reagent containing protease enzyme. Lysis was achieved by incubating the tubes for 20 min at 37 °C followed by 10 min at 95 °C, to complete lysis and deactivate the protease. Subsequently, 30 μL of the lysate was transferred to the PCR tablets, and tubes were placed in the Q7 thermal cycler (Hygiena, Camarillo, CA, USA) for PCR.

### 2.6. Data Analysis

Statistical analyses were performed in R (Version 4.0). For enumeration of Enterobacteriaceae and *E. coli*, a two-way ANOVA was performed to assess the interaction between sampling points (BF, BT, and AT) and treatment groups (no hydrogel beads vs hydrogel beads) during transport. To test the correlation between each sampling point, a pairwise *t*-test was performed, adjusted by the Holm method [28] using the mean values of each sampling point. The effect of the hydrogel beads was evaluated by conducting a pairwise *t*-test adjusted by the Holm method [28] using the means of the treatment and control group. The effect of the treatment and sampling points in *Salmonella* and *E. coli* O157:H7 prevalence was evaluated using a chi-square test for independence. The differences were reported to be statistically significant, considering an alpha value of $\leq 0.05$.

## 3. Results

The study was conducted in the spring, with an average temperature during the days of the experiments of 8.3 °C. Two repetitions of the experiment were conducted on different days. The first experiment included 24 pigs, and the second included 36. Data pertaining to the weight before and after transport, sex, and pen allocation was recorded. Each animal was identified in order to correlate the results within each sample unit (pigs). Preliminary experiments were conducted to prepare hydrogel beads with sensory characteristics that facilitated consumption by the pigs. The results from the preliminary experiments agreed with Aviles-Rosa et al. (2020), who found that pigs preferred feeders sprayed with maternal pheromones [29–31]. These results led to the use of maternal pheromones on the hydrogel beads administered to the animals in the present research. During the transport study, the approximate quantity of beads prepared to supplement the pigs was 30 kg (12 kg for the first 24 pigs, and 18 kg for the second repetition with 36 pigs). No water or food was provided after transport. As indicated, animals were weighed at each sample collection point, and no significant changes in weight were observed.

### 3.1. Enumeration of Enterobacteriaceae and E. coli

Samples were collected at three different time points to assess the concentrations of indicator microorganisms: before fast, before transport, and after transport with the effect of the hydrogel beads. As observed in Figure 1, the concentrations of Enterobacteriaceae in fecal samples BF, BT, and AT were 4.35, 4.66, and 5.09 Log CFU/g, respectively. Enterobacteriaceae counts were significantly higher ($p < 0.05$) at AT compared to BF. As seen in Figure 2, the concentrations of *E. coli* at BF, BT, and AT were 4.03, 4.27, and 4.58 Log CFU/g, respectively. Similar to Enterobacteriaceae, enumeration of *E. coli* showed a significantly higher ($p < 0.05$) concentration at AT than at BF. Enumeration of microorganisms at BF was not significantly different from BT ($p > 0.05$). Similarly, results obtained that BT and AT were not significantly different ($p > 0.05$) between each other.

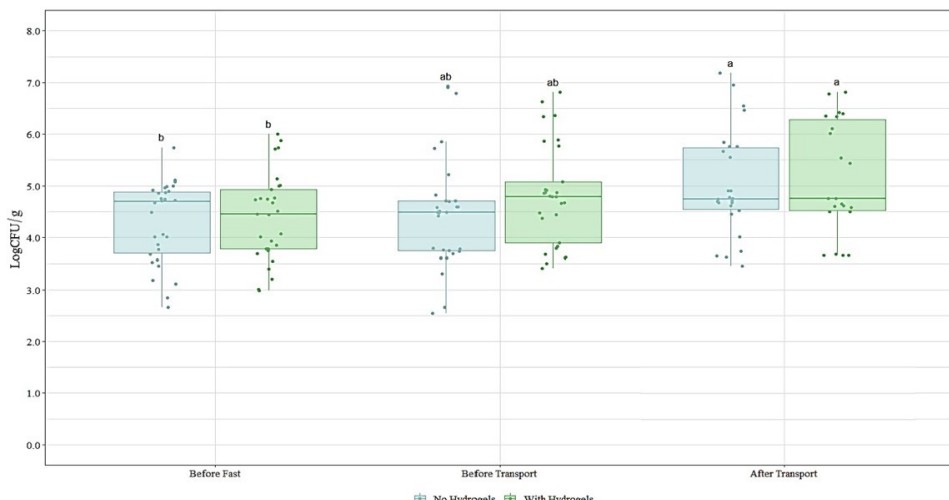

**Figure 1.** Enterobacteriaceae enumeration from fecal samples. Treatment group (*n* = 23) was supplemented with hydrogel beads during transportation (blue bars) and control groups (*n* = 26) did not have access to the beads (green bars). Significant differences (*p* < 0.05) are represented with different letters. Samples were collected at each established point: Before Fast (*n* =59), Before Transport (*n* =60), and After Transport (*n* =49).

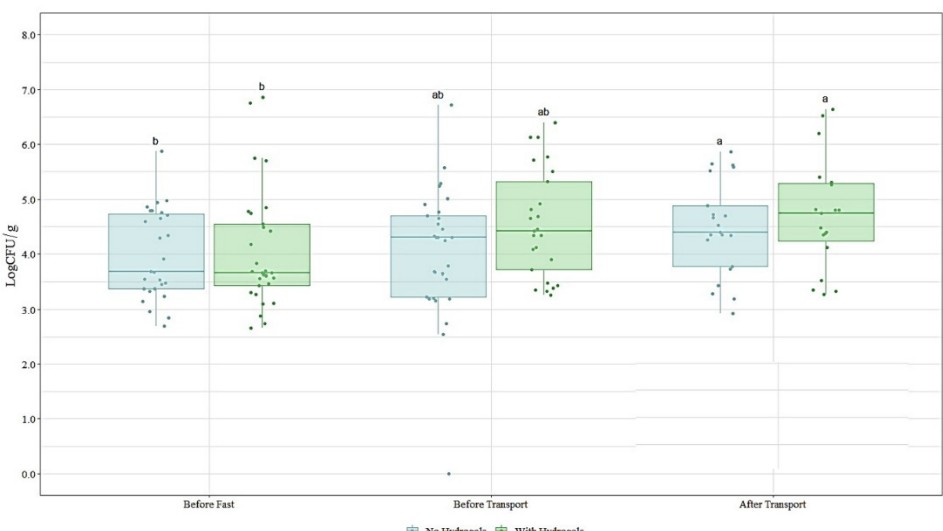

**Figure 2.** *E. coli* enumeration from fecal samples. Treatment group (*n* = 21) was supplemented with hydrogel beads during transportation (blue bars) and control groups (*n* = 21) did not have access to the beads (green bars). Significant differences (*p* < 0.05) are represented with different letters. Samples were collected at each established point: Before Fast (*n* =59), Before Transport (*n* =60), and After Transport (*n* =49).

### 3.2. Detection of Salmonella sp. and E. coli O157:H7

Table 1 summarizes the number and percentage of pigs that were positive for *Salmonella* and *E. coli* O157:H7. A total of 172 samples (BF *n* = 60, BT *n* = 59, and AT *n* = 53) were tested. Of those, 3/172 (1.74%) were positive for *Salmonella* and 89/172 (51.7%) were positive for *E. coli* O157:H7. The prevalence of *Salmonella* positive pigs at BF, BT, and AT was 0.00, 3.39, and 1.89%, respectively (*p* > 0.05). The prevalence of *E. coli* O157:H7 positive pigs at BF, BT, and AT was 53.33, 45.76, and 56.60%, respectively (*p* > 0.05). No significant difference (*p* > 0.05) was observed in the prevalence of both pathogens between the three sampling points.

**Table 1.** Prevalence of *Salmonella* and *E. coli* O157:H7 of market pigs.

| | Before Fast [1] | | Before Transport [1] | | After Transport [1] | |
|---|---|---|---|---|---|---|
| | *Salmonella* | *E. coli* **O157:H7** | *Salmonella* | *E. coli* **O157:H7** | *Salmonella* | *E. coli* **O157:H7** |
| Positive [2] | 0/60 | 32/60 | 2/59 | 27/59 | 1/53 | 30/53 |
| Prevalence | 0% | 53.33% | 3.39% | 45.76% | 1.89% | 56.60% |

[1] Corresponds to the sampling point where the sample was collected. [2] Positive results indicate the total of positives and the total samples analyzed. No difference ($p < 0.05$) was found between the control and the treatment group.

### 3.3. Effect of Alginate Hydrogel Beads

Hydrogel beads were supplemented only during transportation when the animals were fasting. Microbial analyses conducted to establish the concentration of Enterobacteriaceae and *E. coli* (Figure 2) revealed no significant difference ($p > 0.05$) between pigs supplemented with the hydrogels and the control animals (no access to hydrogels). Similarly, the presence of *Salmonella* and *E. coli* O157:H7 was not affected with the supplementation of the beads, and no difference ($p > 0.05$) was observed between the treatment and control groups. As shown in Table 2, *Salmonella* was found in 3.70% and 0.00% in the control and treatment groups, respectively. Alternatively, *E. coli* O157:H7 was present in 51.85% and 61.54% of the animals in the control and treatment groups, respectively ($p > 0.05$).

**Table 2.** Effect of hydrogel on *Salmonella* and *E. coli* O157:H7.

| | Treatment [1] | | Control [2] | |
|---|---|---|---|---|
| | *Salmonella* | *E. coli* **O157:H7** | *Salmonella* | *E. coli* **O157:H7** |
| Positive [3] | 0/26 | 16/26 | 1/27 | 14/27 |
| Prevalence | 0.00% | 61.54% | 3.70% | 51.85% |

[1] Treatment animals supplemented with alginate hydrogel beads during transport. [2] Control samples corresponded to animals with no access to beads during transportation. [3] Positive results indicate the total of positives and the total samples analyzed. No difference ($p < 0.05$) was found between the control and the treatment group.

### 4. Discussion

This study sought to investigate the effects of fasting and transport on microbial load and pathogen shedding in market pigs. As a potential strategy to mitigate bacterial shedding by these food-animals, this research explored the effects of alginate hydrogel beads, supplemented with electrolytes, on the microbial load and pathogen prevalence in fasted pigs during transport. When analyzing bacterial loads recovered AT, an increase in the concentration of Enterobacteriaceae and *E. coli* ($p < 0.05$) in pig's feces was observed relative to the counts obtained BF. However, no changes were observed when comparing the microbial load of BF and BT, nor between BT and AT. Typically for swine, feed is withdrawn 8–24 h prior to slaughter to reduce gut content spillage when the carcass is being eviscerated, which will consequently reduce pathogen contamination of the carcass [32]. The results obtained from the present study indicated that fasting alone does not influence microbial shedding in pigs. However, the pigs used in this study were fasted for only 12 h. One study by Martin-Pelaez et al. (2009) [15], which fasted pigs for 15 h and 30 h, found a tendency for Enterobacteriaceae to increase when feed withdrawal time increased from 15 h to 30 h. To observe the effect of fasting alone, longer fasting periods could be investigated to potentially observe differences in microbial shedding caused by fasting. The total time of feed withdrawal for the pigs in the present study was around 16 h. Since the study by Martin-Pelaez et al. (2009) [15] suggested that the length of fasting increased microbial shedding, perhaps the longer feed withdrawal period could explain why only a significant difference was observed at BF and AT. Other studies have investigated the role of fasting periods on microbial shedding in broilers and cattle. In broilers, the exposure to longer fasting times, up to 24 h, and high temperatures have been known to increase bacterial load in the ileum by up to 1.5 logs [17]. In monogastric animals, feed withdrawal

is correlated with reductions in the quantity and thickness of the mucus in the ileum; this could cause bacterial colonization and growth during fasting periods [18]. In ruminants, fasting reduces volatile fatty acid concentrations, which are known to limit the proliferation of pathogens [33]. If the same is observed in swine, this could provide a reason as to why microbial load increased with fasting and transportation. Further research into mucus thickness and bacterial proliferation would be needed to conclude this in pigs.

With respect to pathogen prevalence, no difference was found among sampling points in *Salmonella* and *E. coli* O157:H7. These results differ from studies indicating that transport can increase prevalence of *Salmonella* by up to 24.5%, which is usually detected at the abattoir after transport [6,34]. Perhaps the increase in prevalence observed in the previous studies can be attributed to the transfer of *Salmonella* between pigs that are in close proximity to those that shed the microorganisms in their feces during transport. It should be noted that transport times vary depending on the proximity of the plant to the slaughter facility. However, there is a twenty-eight-hour law (49 USC, section 80502) that prohibits the confinement of animals beyond a 28 h period [35]. In the present study, a low prevalence of *Salmonella* was found in the farm. The low prevalence of *Salmonella* could indicate superior biosecurity measures, such as good feed practices and vaccination. It is possible that if there had been a higher prevalence of *Salmonella* in the present study, the animals could have transferred the *Salmonella* during transport due to proximity. If this is true, pathogens must be controlled prior to transport to reduce the risks of cross-contamination during transport. With respect to pathogenic *E. coli*, Bach et al. (2004) found that the prevalence of *E. coli* O157:H7 also increased with transport, while Barham et al. (2002) reported that EHEC O157 decreased with transport [36,37].

In other species, fasting and transport have shown to increase the prevalence of pathogens. In cattle, the fasting period can be between 24 h to 48 h prior to transport to slaughter, which can affect the prevalence of *E. coli* O157:H7 and can be an important factor in the internal colonization of pathogens in broilers [38,39].

With respect to the effect of hydrogel beads on microbial shedding, this study showed no significant difference ($p > 0.05$) in the concentrations of Enterobacteriaceae and *E. coli* between the control (no hydrogel beads) and the treatment (hydrogel beads) groups. Additionally, no differences in the prevalence of *Salmonella* and *E. coli* O157:H7 were observed in the animals supplemented with hydrogel beads during transport.

A study conducted to observe the effect of sodium alginate on nutrient digestion in rats found that the short-term effects of consuming hydrogel beads were a gel mass forming in the stomach of the rats [40]. Additionally, alginates can cause enlargement of the stomach and cause satiety, as well as lengthen the gastric emptying time [25]. The results from the study by Guo et al. (2020) [25] could attribute to the issues found with collecting samples from treatment animals after transport in the present study. The main reason hydrogel beads were used in this study was to induce satiety, as well as to provide minerals to the animals. Other studies have focused on hydrogels as delivery systems to induce for other compounds [41–44]. The products of dietary lipids and digestion can produce strong satiety signals. An in vitro study, which investigated release rates of dietary lipids encapsulated in hydrogel beads, shows that 1–50% of the contents were released after 1 h and 20–80% after 2.5 h [41]. Additionally, a study with similar methods found that lipid droplets encapsulated in alginate beads reduced the released free fatty acids. This means that alginate-based hydrogel beads can be potentially used as a model to induce satiety by controlling lipid digestibility [42]. The results from these studies can broaden the field for hydrogel bead use in animals if the focus is to induce satiety. Furthermore, hydrogels have been used as delivery systems for probiotics. A study in which pectin/starch hydrogels were used, showed that the encapsulated cells of *Lactobacillus plantarum* could survive simulated adverse conditions of the gastro-intestinal tract [42]. Moreover, hydrogels made by crosslinking calcium ion and sodium alginate (similar to the ones used in this study) were successful in the oral delivery of *Bifidobacterium breve*. In fact, the viability of *B. breve*

cells was enhanced by the use of hydrogels [44]. These studies show the multi-functional properties of hydrogels, and their many potential uses in the pork industry.

An unexpected outcome of this study was the high prevalence of *E. coli* O157:H7 in the group of animals for this study. This outcome could be attributed to the proximity of the Texas Tech Swine Unit and the Texas Tech Burnett Center Feedlot. Bovine cattle are known to carry members of pathogenic *E. coli*, and these members can be spread by the wind and potentially infect other animals [45]. This could mean that *E. coli* O157:H7 could be spread via cross-contamination. These observations further emphasized the need to implement a One Health approach between the animal production and the food safety industry. By understanding the connection between animal production and food safety, public health can be improved. Further investigations on the role of the proximity between the feedlot and the swine unit, and the role that the environment plays on the spread of pathogens are needed. If the high prevalence of *E. coli* O157:H7 at the swine unit can be correlated to the feedlot, then strategies can be put in place to control the spread of this pathogen.

## 5. Conclusions

The results from this study indicated an increase in the concentration of Enterobacteriaceae and *E. coli* in pigs' feces after transport (AT), compared to the concentrations of both indicator microorganisms at the before fast (BF) stage. These findings indicated that the combination of transportation and fasting could increase the microbial load in the digestive tract. With respect to pathogen prevalence, no difference was found among sampling points in *Salmonella* and *E. coli* O157:H7. This indicated that neither fasting nor transportation affected the shedding of pathogens in swine feces. The contrast between the studies above and the present study could potentially be attributed to different sampling locations and sampling methods, populations of *Salmonella* and *E. coli* O157:H7, and different fasting and transport times. Although our results showed no significant differences in the prevalence of the observed pathogens due to fasting and transport, other studies have shown a negative effect of fasting and transport in pathogen shedding. As a result, it continues to be an important research area in the food safety field.

Additionally, no differences in the concentrations of indicator microorganisms and pathogens were observed between the control and treatment group. This indicated that hydrogel bead supplementation did not affect the shedding of indicator microorganisms and the prevalence of pathogen shedding. Research on hydrogel beads as delivery agents in animals is limited. Therefore, these results implicated the need for additional experiments, in which longer exposure times to hydrogel beads is needed, in order to further assess the effects of supplementation.

Potential areas of study to complement research of this nature would include quantitative measures of animal stress to correlate with microbial results. Since stress has been linked with pathogen shedding and colonization, using hydrogel beads as a supplement to reduce stress in the animal could achieve a greater degree of food safety.

Although the present study did not observe any positive effect of hydrogel bead supplementation in swine during transport, hydrogel beads continue to be an important area of research. There are multiple studies that have sought out to control pathogen shedding in animals to reduce the entrance of these microorganisms into the supply chain. However, given the harsh environment of the gastrointestinal tract, it is necessary to protect what is being supplemented so that it can have a greater impact. Thus, the use of hydrogel beads to encapsulate what is being supplemented is an area of study that has a potential to make a positive impact in the food supply chain. By using hydrogels to protect what is supplemented, the release of those materials into the gastrointestinal tract can be optimized. Through studies like these, we hope to gain more information about the uses of hydrogel beads in the food industry. More research is needed regarding the metabolic responses that the body has to the hydrogel beads and the relation to bacterial shedding.

**Author Contributions:** Conceptualization, A.C. and A.G.; methodology, A.C. and A.G.; formal analysis, M.F. and D.A.V.; investigation, M.F.; resources, A.C.; writing—original draft preparation, M.F.; writing—review and editing, A.C. All authors have read and agreed to the published version of the manuscript.

**Funding:** This research received no external funding.

**Institutional Review Board Statement:** This experimental protocol was approved by the Institutional Animal Care and Use Committee (IACUC) and the animal handling followed the guidelines of the IACUC in compliance with the Association for the Assessment and Accreditation of Laboratory Animal Care, International (Protocol:19104-12).

**Informed Consent Statement:** Not applicable.

**Data Availability Statement:** Project data has been reported within this manuscript.

**Acknowledgments:** The authors would like to acknowledge the Texas Tech New Deal Farm Swine Unit and ICFIE Lab at Texas Tech University who were instrumental in the development of this research.

**Conflicts of Interest:** The authors declare no conflict of interest.

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
