# Peer review of "Dynamics of Microbial Shedding in Market Pigs during Fasting and the Influence of Alginate Hydrogel Bead Supplementation during Transportation"

_2036-7481, doi:10.3390/microbiolres12040065_

Round 1

Reviewer 1 Report

The manuscript has been prepared carefully and reasonably well written, but needs some comments:

- the title is not informative in its current form, because it should also indicate that the studied dynamics of microbial shedding was analyzed in terms of the additional factor (supply of hydrogel beads)

 - low percentage of Salmonella isolates (1 or 2 or no strains) does not warrant statistical analysis. In this case, the related limitations should be indicated

- Another issue that should be noted in the discussion is the limitations related to the short test time (12 hours of fasting and 4 hours of transport), which could have resulted in the lack of differences in the comparative statistical analysis. It should also be noted how many hours, under standard conditions, it takes the transport to the slaughterhouse and the preceding fasting, so that the test results could possibly have a practical impact, i.e. whether the supply of hydrogel beads really prevents increased excretion / multiplication of pathogens and thus an increased risk of contamination

Author Response

Point 1:

The title is not informative in its current form, because it should also indicate that the studied dynamics of microbial shedding was analyzed in terms of the additional factor (supply of hydrogel beads)

Response 1:

Thank you for you recommendation, we have changed the title of the manuscript and it now reads:

“Dynamics of Microbial Shedding in Market Pigs During Fasting and Influence of Alginate Hydrogel Bead Supplementation During Transportation”.

Point 2:

Low percentage of Salmonella isolates (1 or 2 or no strains) does not warrant statistical analysis. In this case, the related limitations should be indicated

Response 2:

In this study, we conducted two type of testing. One was concentration of indicator organisms and the second was prevalence of pathogens. For the second part, we evaluated the number of animals that carried Salmonella and E. coli O157:H7 (as per the PCR test), at each of the sampling points. Since you indicate on your fisrt line “Percentage of Salmonella isolates”, we would like to clarify that Salmonella and E. coli O157:H7 were not isolated.  On the other hand, when evaluating the prevalence of these two pathogens, we did find a very low number for Salmonella (between 0 to 3.4% prevalence). Regardless of having low values, we considered important to report them as they were our findings. We consulted with an statistitian the if it was possible to compare the number of animals that were Salmonella or E. coli O157 positive on each sampling point and conduct statistical analysis.  We consulted again after the revisions to make sure is appropriate to report as we did.  Thus, we would like to keep the data as reported in the manuscript. 

If there was any specific recommendation we are open to any suggestions.

Point 3:

Another issue that should be noted in the discussion is the limitations related to the short test time (12 hours of fasting and 4 hours of transport), which could have resulted in the lack of differences in the comparative statistical analysis. It should also be noted how many hours, under standard conditions, it takes the transport to the slaughterhouse and the preceding fasting, so that the test results could possibly have a practical impact, i.e. whether the supply of hydrogel beads really prevents increased excretion / multiplication of pathogens and thus an increased risk of contamination

Response 3

With respect to the time selected for fasting and transportation, in swine, feed is typically removed from 8-24h before slaughter. To clarify this, we have added a paragraph in the Discussion that states: “Typically for swine, feed is withdrawn 8-24 h prior to slaughter to reduce gut content spillage when the carcass is being eviscerated, which will consequently reduce pathogen contamination of the carcass [32].”. Reference 32 was also added to support the statement.   Consistently with your comment, we found studies in which they experimented with extended fasting periods and they observed changes in microbial concentrations. We have several citations in this regard. In response to your recommendation of noting limitations pertaining shorter fasting times, we stated in our discussion that the short fasting period could be a reason why we did not see any significant differences in indicator concentrations. 

To explicitly show how this was addressed, we have included a paragraph in the discussion that states: “The results obtained from the present study indicate that fasting alone does not influence microbial shedding in pigs. However, the pigs used in this study were fasted for only 12h. One study by Martin-Pelaez et al., (2009) [15] which fasted pigs for 15h and 30h, found a tendency for Enterobacteriaceae to increase when feed withdrawal time increased from 15h to 30h. To observe the effect of fasting alone, longer fasting periods could be investigated to potentially observe differences in microbial shedding caused by fasting.”

With respect to transportation, no standard times exist as it depends on the distance bewtween the farm and the slaughtering facility. We could have elected to investigate long or short transportation times; nonetheless, this project was linked to another research pertaining animal welfare, in which short times were tested as they may have an impact in animal stress. Animal welfare data is not part of the present study.  Having said that, there is a regulation indicating that animals should not be transported for more than 28h. A new paragraph was added in our discussion that states: “It should be noted that transport times vary depending on the proximity of the plant to the slaughter facility. However, there is a twenty-eight-hour law (49 USC, section 80502) that prohibits the confinement of animals beyond a 28h period”. Anew citation was also added.  

It is noted in the manuscript that differences in results from past studies and the present study could be due to transportation times. In our discussion there is a parapgraph that indicates:  The contrast between the studies above and the present study, could potentially be attributed to different sampling locations and sampling methods, populations of Salmonella and E. coli O157:H7 and different fasting and transport times.”

Finally, we respect to the effect of hydrogel beads, we evaluated changes in excretion as you mentioned, but we did not test the risk of contamination.  We found no changes in microbial shedding after the supplementation with the hydrogel beads.

Additional note to the reviewer:

The manuscript has been reviewed for spelling mistakes which have been corrected and can be found in document.

Reviewer 2 Report

In affiliation of authors replace 2 as 1, firstly is affiliation of first author.

In the abstract family is without italics, please correct in all text. Family without italics is not correct for several years. . Author mentioned Enterobacteriaceae and E. coli is not from this family?

Pigs were classified into three weight categories, please specify how many animals were into three groups. And how old were the animals in each group.

The manuscript is not very good quality for the presentation of results.

In this manuscript is missing novelty of  study, about this topic exist several manuscripts.

Manuscript has many shortcomings.

Conclusion of study is missing.

Author Response

Point 1:

In affiliation of authors replace 2 as 1, firstly is affiliation of first author.

Response 1:

Affiliation has been corrected based on order as recommended.

Point 2:

In the abstract family is without italics, please correct in all text. Family without italics is not correct for several years.

Response 2:

We have italicized the family name of Enterobacteriaceae in the text as it is indicated to do so in:  https://wwwnc.cdc.gov/eid/page/scientific-nomenclature. Could the reviewer please specify if the comment is saying that “Enterobacteriaceae” should not be italicized? In other studies, conducted recently, the name of Enterobacteriaceae is italicized in the manuscript (Janda & Abbot, 2021; Sargun et al., 2020). If there is a reason that we are not aware as to why it should not be italicized, we are open to review.

Point 3:

Author mentioned Enterobacteriaceae and E. coli is not from this family?

Response 3:

We are not completely sure what the recommendation is or how to address this comment. We read our paper again and did not find any statement made where we mentioned “Enterobacteriaceae and E. coli is not from this family

  1. coli is a member of the family Enterobacteriaceae. We tested for Enterobacteriaceae and E. coli, as is well known, that different types of media can be used to enumerate each group of organisms separately. In other words, is fairly common to report counts Enterobacteriaceae and separately counts of E. coli using different media, and those two are frequently used as indicators in the food industry.

Since pigs are food aninmals, we elected to use Enterobacteriaceae and E. coli counts as indicators.  E. coli will serve as an indicator or possible presence of pathogenic E.coli and how the population would change with transportation; whereas Enterobacteriaceae counts, is a broader indicator of potential changes in microbial population representing any member of this family. 

Point 4:

Pigs were classified into three weight categories, please specify how many animals were into three groups. And how old were the animals in each group.

Response 4:

To address this comment, we have added in the methods section a clarification, and the paragraph now reads “ Prior to the study, pigs were classified in equal numbers (20 animals/category) into three different weight categories: light (89-99kg), medium (100-105kg), and heavy (106-116kg).”

Point 5:

The manuscript is not very good quality for the presentation of results.

Response 5:

We appreciate the comment; however, having a specific recommendation will help us understand specifically where the reviewer found flaws in the presentation of results.

In the case of enumeration, results of this study are presented in a way in which the reader can visualize the concentrations of the indicator microorganisms. The box plots show the three different sampling points (BF, BT, AT) on the x-axis and the control and treatment groups in each time point. The mean of each sampling point was used to determine a significant difference in indicator organisms due to fasting and transport. The mean of each treatment group was used to determine the effect of the hydrogel beads. By combining both sampling point and treatment groups in one figure, the reader can easily visualize the results. In the case of the detection, prevalence studies are usually presented in a table. Thus, we saw fit to create two tables. One to visualize the prevalence of both E. coli O157:H7 and Salmonella with fasting and transport, and the other for the effect of the hydrogel beads. We are open to any specific comment the reviewer has about the results.

We reviwed the paper again; but not knowing what to change made it difficult for us to identify the areas for improvement. We are open to receive feedback in this regard.

Point 6:

In this manuscript is missing novelty of study, about this topic exist several manuscripts.

Response 6:

We appreciate the observation; however, it was not possible to make changes to respond to this comment since it was not a specific aspect to address.

Interventions to control pathogen shedding in animals and hydrogels have been studied rather extensively in the past in a variety of ways. However, using hydrogels as a delivery system to control pathogens in is not a topic that is very extensively researched. Through this and other studies, we hope to obtain more information on the potential uses of hydrogel beads as delivery agents in the food supply chain.

Point 7:

Manuscript has many shortcomings.

Response 7:

We have added some more information to the discussion section in order to improve the manuscript. Several improvements made to the paper can be seen in the document since we use track-changes feature.

Unfortunately, the comment is not specific, we can’t decide what parts should be modified based on this observation.  Any specific recommendations regarding the shortcomings of the manuscript are welcomed and will be taken into consideration.

Point 8:

The manuscript has no conclusion.

Response 8:

In the template for this journal a conclusion is not required; because of this, we have concluded the paper in the discussion section.  Nonetheless, we appreciate the observation and made a few changes to make conclusions more evident.   

The following paragraphs in quotation marks were improved in our discussion section to address this comment. These are examples where we presented conclusions. Please keep in mind that we are pasting paragraphs to show how we addresseed the comment, but these paragraphs are just portions of the manuscript; thus, you may need to refer the the paper to read how this flows into the discussion.

“When analyzing bacterial loads recovered AT, an increase in the concentration of Enterobacteriaceae and E. coli (P<0.05) in pig’s feces was observed relative to the counts obtained BF. However, no changes were observed when compared microbial load of BF and BT, neither between BT nor AT. These findings indicate that the combination of transportation and fasting can increase the microbial load in the digestive tract.”

“The results obtained from the present study indicate that fasting alone does not influence microbial shedding in pigs.”

“With respect to pathogen prevalence, no difference was found among sampling points  in Salmonella and E. coli O157:H7. This indicates that neither fasting nor trans-portation do not affect the shedding of pathogens in swine feces.”

“The contrast between the studies above and the present study, could potentially be at-tributed to different sampling locations and sampling methods, populations of Salmonella and E. coli O157:H7 and different fasting and transport times.”

“Although our results show no significant differences in the prevalence of the observed pathogens due to fasting and transport, other studies have shown a negative effect of fasting and transport in pathogen shedding. Because of this, it continues to be an im-portant research area in the food safety field.”

“Additionally, no differences in the prevalence of Salmonella and E. coli O157:H7 were observed in the animals supplemented with hydrogel beads during transport. This in-dicates that hydrogel bead supplementation does not affect the shedding of indicator microorganisms and the prevalence of pathogen shedding. Research on hydrogel beads as delivery agents in animals is limited. Therefore, these results implicate the need for additional experiments in which longer exposure times to hydrogel beads in order to further assess the effects of supplementation.”

“These observations further emphasize the need to implement a One Health approach between the animal production and the food safety industry. By understanding the connection between animal production and food safety, public health can be improved. Further investigations on the role of the proximity between the feedlot and the swine unit and the role that the environment plays on the spread of pathogens are needed. If the high prevalence of E. coli O157:H7 at swine unit can be correlated to the feedlot, then strategies can be put in place to control the spread of this pathogen.”

“Potential areas of study to complement research of this nature is to include quanti-tative measures of animal stress to correlate with microbial results. Since stress has been linked with pathogen shedding and colonization, by using hydrogel beads as a sup-plement to reduce stress in the animal, a greater degree of food safety could be achieved.”

Note to the reviewer:

The manuscript has been reviewed for spelling mistakes, which have been corrected and can be found in document.

Round 2

Reviewer 2 Report

Point 2: I m not agree with response

Please check the web page: https://wwwnc.cdc.gov/eid/page/scientific-nomenclature.

Last correction is from 2014 year. I can send you several manuscripts with Enterobacteriaceae without italic.

Point 3: Please check line 78-79. It was just a question, why are you writing Enterobacteriaceae and separate E. coli?

Point 6: It was the reviewer comment. In this age it is very interesting to publish new results or indicate the novelty of a manuscript.

Point 7. I don't understand the answer, Just check the manuscript and correct all the mistakes.

Point 8 is not mandatory, but for authors is not mandatory reviewer recommendation?

Author Response

Point 2: I m not agree with response. Please check the web page: https://wwwnc.cdc.gov/eid/page/scientific-nomenclature.

Last correction is from 2014 year. I can send you several manuscripts with Enterobacteriaceae without italic.

Response to Point 2

Changes were made as recommended. We have reviewed the recommended text and change all Enterobacteriaceae to spell them in the manuscript without italics.

Point 3: Please check line 78-79. It was just a question, why are you writing Enterobacteriaceae and separate E. coli?

Response to Point 3

We have written Enterobacteriaceae separate from E. coli because two different tests were used and counts were reported separately. We used 3M petrifilm Enterobacteriaceae (https://www.3m.com/3M/en_US/p/d/v000530644/) and also 3M petrifilm E. coli/coliforms and reported only E. coli (https://www.3m.com/3M/en_US/p/d/b00013933/).

Point 6: It was the reviewer comment. In this age it is very interesting to publish new results or indicate the novelty of a manuscript.

Response to point 6

We appreciate the clarification and the recommendation.  To address this, we have added a paragraph in the conclusions section, which represents our view of the novelty of this research. This paragraph reads:

Although the present study did not observe any positive effect of hydrogel bead supplementation in swine during transport, hydrogel beads continues to be an important area of research. There are multiple studies which have sought out to control pathogen shedding in animals to reduce the entrance of these microorganisms into the supply chain. However, given the harsh environment of the gastrointestinal tract, it is necessary to protect what is being supplemented so that it can have a greater impact. Thus, the use of hydrogel beads to encapsulate what is being supplemented is an area of study that has a potential to make a positive impact in the food supply chain. By using hydrogels to protect what is supplemented, the release of those materials into the gastrointestinal tract can be optimized. Through studies like these, we hope to gain more information about the uses of hydrogel beads in the food industry”

Point 7. I don't understand the answer, Just check the manuscript and correct all the mistakes.

Response to point 7

Thank you for the clarification. We have reviewed the manuscript and corrected all mistakes/spelling errors found. Please refer to the manuscript to observe our changes and improvements.

Point 8 is not mandatory, but for authors is not mandatory reviewer recommendation?

Response to point 8

We have addressed your recommendation by separating Conclusions and Discussion.  Thus, the manuscript now has a “Conclusions” section. Please refer to the paper to find the changes.  
